# Uncertainty estimation in classification via weighted test-time augmentation

## Abstract

In classification, deep learning models are considered superior in terms of prediction accuracy compared to standard statistical models. However, these models are often overconfident in their predictions, which inhibits their use in safety critical applications where mistakes can lead to disastrous consequences. To address this issue, several uncertainty quantification methods have been proposed to introduce more reliable predictions and better calibration. In this paper, we focus on a universal uncertainty quantification method called test-time augmentation (TTA). We then present a weighted version of test-time augmentation (WTTA) that introduces weights to the algorithm to generate better augmentations. Our approach is then illustrated with various models and data sets. In a simulation study, we show that the WTTA approach produces better uncertainty estimates as we are able to compare to the real uncertainties in the data. Furthermore, the method is applied to two benchmark data sets used in the development of machine learning models. Although, it is a rather simple post-process method, WTTA is arguably able to outperform the standard TTA and temperature scaling methods in terms of calibration error and prediction accuracy, especially in small data sets.

## 1 Introduction

Deep learning models are more popular than ever in many fields of science, and for good reason: the prediction accuracy of these high complexity models is unmatched. However, in many fields such as medical imaging, the ability to understand the decision made by this type of black-box model may be the most essential part for the expert (Nair et al., 2020). This lack of trust in the model and its predictions is a serious limiting factor in high-risk application domains (for examples, see Molnar (2022)).

*Uncertainty quantification* (UQ) is a crucial part of the analysis if the decision made by the model is critical (for a comprehensive review, see Abdar et al. (2021)). Without UQ, the predictions are typically overconfident (Guo et al., 2018). There exist various different ways of introducing uncertainty estimation into the analysis, for example by using additional models especially for uncertainty, by transforming the original (deterministic) model into stochastic or by combining several different models at inference (Gawlikowski et al., 2023). While these types of approaches yield good results, their implementation is highly dependent on the model at hand, and thus they lack a general method for estimating uncertainty.

Typically, UQ methods in deep learning can be divided into four subcategories based on their way of managing the uncertainty of the model. Single network deterministic methods, Bayesian methods, Ensemble methods, and Test-time augmentation methods Gawlikowski et al. (2023). Although comparisons between all subcategories could be made, it might not be sensible as the methods differ greatly in terms of model construction, memory consumption, and implementation. Here, we focus on post-process methods, meaning that they can be implemented after the initial model is trained.

Instead of creating additional deep learning models or transforming existing ones, one option is to alternate input data with so-called *Test-time augmentation* (TTA) to quantify the uncertainty of the predictions (Kimura, 2021). TTA is based on the idea that we can generate multiple predictions for a single observation by alternating the input data in a reasonable way (Gawlikowski et al., 2023). In other words, these augmented

data points enables the exploration of the space of possible predictions for that observation, thus capturing the uncertainty of the prediction.

In this paper, we propose *weighted test-time augmentation* (WTTA) as a post-processing step for a standard statistical analysis with various models and data sets. While TTA certainly brings many advantages to the analysis, there still isn't a consensus on how accurately the method quantifies the uncertainty. Moreover, the impact of different properties of the data on the augmented predictions, for example skewness, measure scale, or model complexity, is rather unknown. Our aim is to tackle these challenges by first creating controlled data sets to explore the data uncertainty and its quantification. Second, various different models and real-life data are analyzed to show the practicality of TTA and WTTA for different fields of study.

The novelties of the article are the following:

- We introduce WTTA, a general model-agnostic method that can be implemented with ease for post-process analysis without requiring original data for model fitting.

- The WTTA method improves the uncertainty measure of the prediction. This is shown with simulated data where comparisons to the known uncertainty distribution can be made.

- WTTA is illustrated with both tabular and image data as well as with different machine learning models, such as neural networks and random forests.

- We show that, in addition to providing uncertainty quantification, WTTA also improves prediction accuracy.

- WTTA with weighted means is shown to be a robust method for out-of-distribution samples.

All the scenarios are written with R and they can be found in Github [1]. In Section 2, we go through the related work. In Section 3, the WTTA method is described for both tabular and image data. In Section 4, we analyze several simulated data sets, where the underlying uncertainty structures are known. In Section 4.2, we analyze a real-life tabular data set of wine classification (Aeberhard & Forina, 1992), and in Section 4.3, we study image classification with the MNIST data set (Deng, 2012). In Section 5, we provide some closing remarks.

## 2    Related work

TTA is widely used in image analysis because there is a natural connection between image transformations, such as flipping, cropping and turning, and data augmentation (Ayhan & Berens, 2022; Lyzhov et al., 2020). It can be said that the roots of the TTA approach are in medical image analysis (Ronneberger et al., 2015). One common application for TTA is in segmentation analysis, where the general aim is to find the edges of certain images within images (Nalepa et al., 2020; Wang et al., 2019a;b). One of the most recent work on TTA was written by Hekler et al. (2023), where the authors implemented and compared a traditional TTA approach to other post-hoc calibration methods, showing that TTA results in better calibration. Furthermore, TTA can be used to estimate confidence in a classifier's predictions (Bahat & Shakhnarovich, 2020).

The selection of the augmentations is not self-explanatory and requires attention to various factors of the problem at hand (Shanmugam et al., 2021). Moshkov et al. (2020) utilized TTA in segmentation analysis in conjunction with majority voting to create the final output segmentation mask. Lyzhov et al. (2020) introduced a greedy policy search to learn augmentation policies, that improved robustness and provided better uncertainty estimation. Kim et al. (2020) proposed a method for learning a loss predictor from the training data for instance-aware test-time augmentation selection.

Interestingly, TTA is utilized not only to capture uncertainty, but also to improve the accuracy of the predictions (Moshkov et al., 2020). There have been many experiments, with various models and data sets, which have shown improved accuracy (Shanmugam et al., 2021). However, while the net accuracy is generally

---

[1]https://anonymous.4open.science/r/WTTA_uncertainty_estimation-A4EC/

improved, there is a chance that correct predictions can flip incorrectly, which depends on the nature of the problem, the size of the training data, and other factors (Shanmugam et al., 2021).

Several calibration methods for deep learning models have been studied to tackle the common problem of overconfidence (for a recent survey, see Wang (2024)). Calibration is especially necessary in safety critical applications such as medical imaging (Caruana et al., 2015). Common post-process methods for calibration include *histogram binning* (Zadrozny & Elkan, 2001), *isotonic regression* (Zadrozny & Elkan, 2002) and *Platt scaling*, which includes *temperature scaling* (Platt et al., 1999).

With TTA, Ayhan & Berens (2022) were able to produce well calibrated probabilistic outputs for state-of-the-art neural networks. Ashukha et al. (2020) further showed that TTA improves calibration to in-distribution data and enhances detection of out-of-distribution data.

## 3   Method

The basic idea behind TTA is to create and utilize multiple samples from a single observation in order to measure predictive uncertainty. These new samples are generated by augmenting the original data point. With these augmented samples, it is possible to capture the uncertainty as they can be viewed as different realizations of the original. The general procedure of TTA is visualized in Figure 1.

In practice, TTA is a relatively simple method, as it is independent of the model type. Furthermore, it does not require original or additional data, and the underlying model remains unaltered. However, there are some common pitfalls in the implementation of TTA. First, the augmented samples have to represent the data distribution well. Second, if the augmented samples differ greatly from the original data point, the method can generate extroneous noise to the predictions, thus lowering the predictive accuracy.

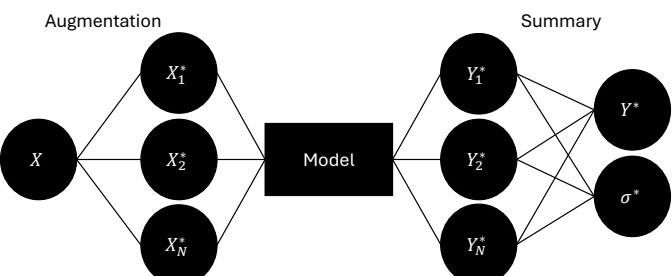

Figure 1: A visualization of test-time augmentation method for single observation $X_i$. First, augmentations of the observation $X_1^*, X_2^*, \ldots, X_N^*$ are generated from a specified augmentation distribution. Next, the augmentations are used as inputs for the pre-trained model and outputs $Y_1^*, Y_2^*, \ldots, Y_N^*$ are received. Finally, summary statistics of the output distribution are evaluated.

### 3.1   TTA with tabular data

In TTA, the augmentation of the data is carried out after the training of the model. With tabular data, the choices for the augmentation distributions are highly dependent on the variables' distributions. As TTA estimates the aleatoric uncertainty of the data point, the chosen distribution or function must closely represent the data in such a way that out-of-distribution points will not be generated (Ayhan & Berens, 2022).

Let $X_i \in \mathbf{R}^m$ represent the $i$th input vector with dimension $m$ and let $Y_i \in [0,1]$ be the corresponding classification output from a pre-trained posterior model $p(Y|X, \theta)$. Augmentations of the original inputs $X_{i,j}^*$, $j = 1, 2, \ldots, N_{aug}$, where $N_{aug}$ is the number of augmentations per input, are generated from an augmentation distribution,

$$X_{i,j}^* \sim G(X_i, t),$$

where $t \in \mathbf{R}^p$ is a set of parameters for the augmentation. One fairly simple way to implement the TTA method is to generate the augmentations from a Gaussian distribution $N(\mu, \sigma I)$ such that the mean vector

$\mu \in \mathbf{R}^m$ is set as the data point under examination, $\mu = X_i$. The new set of augmented observations are thus randomly generated via

$$X_{i,j}^* \sim \mathrm{N}(X_i, \sigma I).$$

The augmented observations $X_{i,j}^*$ are then used as inputs in the pre-trained model to evaluate the outputs $Y_{i,j}^*$. The final prediction is then combined from these augmented predictions, typically by taking their mean. However, experimental results have shown that the simple average may not be the optimal approach for predictive accuracy (Shanmugam et al., 2021). In this paper, the augmented predictions are combined with the weighted mean

$$\bar{Y}_i^* = \frac{1}{N_{\mathrm{aug}}} \sum_{j=1}^{N_{\mathrm{aug}}} w_j Y_{i,j}^*,$$

where the weights $w_j = \frac{1}{||X_i - X_{i,j}^*||^2}$, are the inverses of the squared Euclidean distances to the original data point $X_i$.

Furthermore, the augmented predictions can be investigated with other sample statistics. In this paper, the sample variance $S^2$ is used to find the edge of the decision plane.

### 3.2 TTA with image classification

When analyzing image data, such as the MNIST or the ImageNet data sets, the simple approach of augmenting each single pixel value separately is insufficient. Neighboring pixels are highly dependent and altering their values without considering this correlation produces unrealistic realizations of the data. With TTA, the augmented observations should resemble the data (Ayhan & Berens, 2022).

Fortunately, in the field of image classification, various different types of image transformations have been utilized over the years. In the most typical cases, flipping, cropping and scaling are a convenient way to generate augmentations (see eg. (Shanmugam et al., 2021)). An important aspect to consider is how much the images can be reasonably transformed. For example, in digit classification, digits cannot be rotated too much without mixing up different classes (e.g., "6" and "9").

## 4 Experiments

The WTTA approach is next studied with three different experiments to highlight the performance of the method with varying models and data types. A quick summary of each experiment is listed below.

1. A simulation study: WTTA is studied with simulated data in two scenarios, a simpler and a more complex one, so that the augmentation uncertainty can be compared with the known variation in data generation.

2. A wine classification study with multinomial logistic regression: WTTA is applied to a simple real-life data set with a relatively low number of observations.

3. An image classification study with neural networks: This study considers image classification with the benchmark MNIST data set to highlight the benefit of using WTTA with neural networks, showing the range of models the method is capable of managing.

### 4.1 A simulation study with random forests

In this study, we generate a simulated data set to present the advantages of the WTTA approach. The main benefit of utilizing fully controlled simulated data is that the the data generation method is known and comparisons to this otherwise hidden information can therefore be made directly. This is especially beneficial when examining uncertainty which is generally unattainable with real-life data.

Next, we carry out two simulation scenarios:

- A simple data set where $X$ is an $n \times 2$ matrix of the simulated values of two explanatory variables and $Y$ is an $n \times 1$ vector of binary outputs. An $n \times 1$ vector $P$ of probability values represents the variation in the generation of the output $Y$.

- A more complex data set where $X$ is an $n \times 5$ matrix. The output vector $Y$ is similar to the output in the previous scenario.

In both scenarios, a total of 1500 observations are simulated. These are divided equally into training, validation, and test data sets.

---

**Algorithm 1** Data simulation

---

1: **procedure** SIM($n$, $X_f$, $\sigma$, $seed$)
2:     $set.seed(seed)$
3:     Generate $X_1 \sim \text{Unif}(0, 1, n)$ and $X_2 \sim \text{Unif}(0, 1, n)$
4:     Create the $n \times 2$ matrix $X = \begin{bmatrix} X_1 & X_2 \end{bmatrix}$ with rows $X_{(i)}$
5:     **for** $i = 1$ to $n$ **do**
6:         Calculate the squared distance $d_i = \min(||X_f - X_{(i)}||^2)$
7:         Calculate the probability $p_i = \exp(-\frac{d_i}{2\sigma})$
8:         Generate output $y_i \sim \text{Bern}(p_i)$
9:     **end for**
10:    Set $Y = \{y_i\}$ and $P = \{p_i\}$.
11:    **return** $X, Y, P$
12: **end procedure**

---

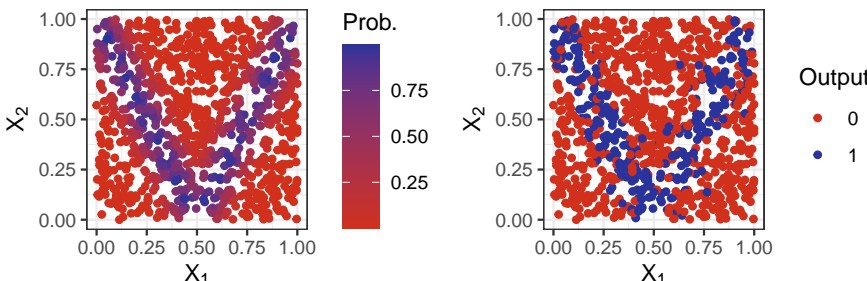

Figure 2: One example data set ($n = 1000$) simulated with Algorithm 1 using the function $f(x) = 2|x - 0.5|$ to create the V-shape $X_f = \{(x, f(x)) \mid 0 \leq x \leq 1\}$ in the generation of the output $Y$. The probabilities $P$ are shown on the left panel and the final generated output values $Y$ are shown on the right panel.

### 4.1.1 Analysis with two-dimensional input

First we analyze data with only two explaining variables. The motivation for this simple scenario is that the data and the WTTA process can be visualized in a plane. In Figure 3, top left panel, the probabilities of generating the output value 1 for each data point are visualized, indicating a V-shape in the plane. Note that these probabilities correspond to the true uncertainty of the points since the output variable $Y$ is then generated based on these probabilities.

First, in total of 4 types of augmentations is implemented, each with different value for parameter $\hat{\sigma}$: $\hat{\sigma} = 1$, $\hat{\sigma} = 0.1$, $\hat{\sigma} = 0.01$ and $\hat{\sigma} = 0.001$. The impact of the value is clearly visualized in Figure 3 middle and bottom row, where the color of the point refers to the mean over the augmented predictions. Generally, the higher the value of $\hat{\sigma}$, the more blurry the predictions become. These can further be compared to the original predictions (Figure 3 top right) where the predictions are made without any augmentation to the data points. Without

any additional information, it seems that augmentations with $\hat{\sigma} = 0.01$ and $\hat{\sigma} = 0.001$ (Figure 3 bottom row) are clearly superior to the other augmentations, as they are more similar to the real probabilities behind the data.

Next, the optimal value for parameter $\hat{\sigma}$ is searched over a discrete grid. In Figure 4 the value of mean sum of squares error is evaluated with varying values for $\hat{\sigma}$. In this instance, the lowest value is achieved when $\hat{\sigma} = 0.03$. Furthermore, we can clearly see that augmentation improves the total MSE of the model when parameter $\hat{\sigma}$ is set $\hat{\sigma} < 0.08$.

With the optimal value for parameter $\hat{\sigma}$, the final comparison between the original model and WTTA can be made. As was already mentioned, WTTA improves the MSE from 0.0802 to 0.0778, the method also improves the uncertainty error of the model from 0.0172 to 0.0118 (Table 1). Note that the uncertainty error in Table 1 is evaluated by comparing the true underlying probability of belonging to class 1 to its estimated value.

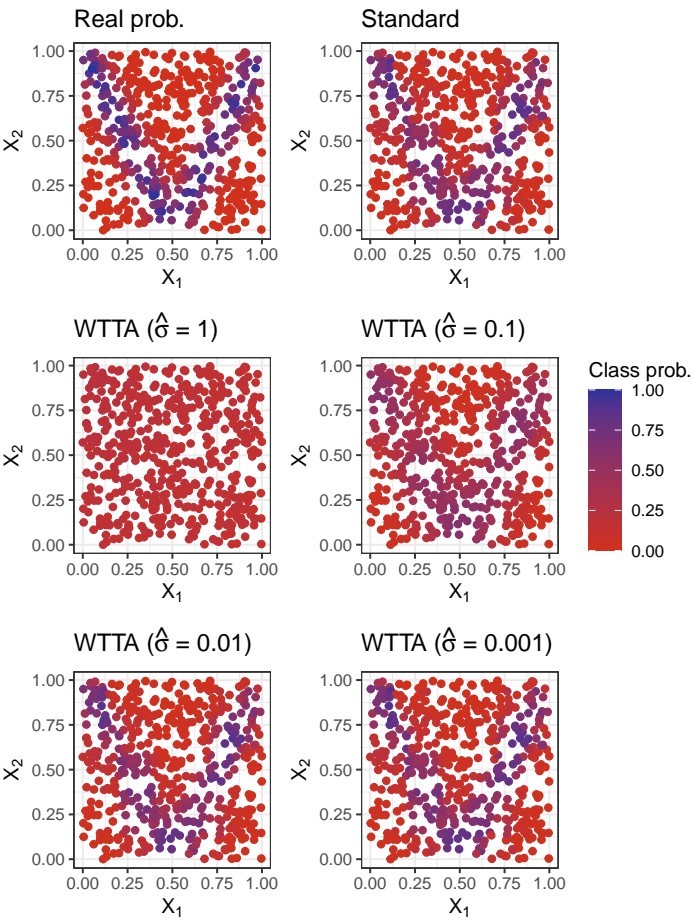

Figure 3: The true probabilities and the corresponding predicted values in a single example data set ($n = 500$) with two-dimensional input. Original predictions refer to the predictions made by the RF-model and augmented predictions refer to the mean of the WTTA predictions, with varying values of parameter $\hat{\sigma}$.

### 4.1.2 Analysis with five-dimensional input

Next we analyze a more complex scenario where five explatory variables are simulated from distributions with varying skewness. The sampling of the first two variables is achieved with an asymmetric Laplace distribution

|  | Random forest | Random forest + WTTA |
|---|---|---|
| Prediction error | 0.0802 | 0.0778 |
| MSE | 0.0172 | 0.0118 |

Table 1: Performance of random forest and random forest with WTTA in the simulation scenario with two-dimensional input.

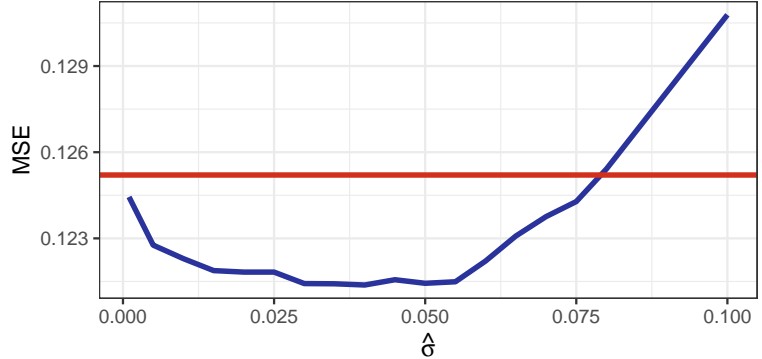

Figure 4: The MSE values with varying values of parameter $\hat{\sigma}$ in the simulation scenario with two-dimensional input. The red line refers to the prediction error without WTTA.

with skew parameter $\kappa = 0.3$, using the r-package *LaplacesDemon* Statisticat & LLC. (2021). The last three variables are simulated from a uniform distribution.

Similarly to the previous example, a single scale parameter $\hat{\sigma}$ needs to be determined for the augmentation. Once again the optimal value for $\hat{\sigma}$ is sought by running the augmentation with varying $\hat{\sigma}$ and evaluating the sum of squares error. In Figure 5, we observe that the minimizer is at $\hat{\sigma} = 0.3$. Furthermore, WTTA with this optimal value for $\hat{\sigma}$ improves both the MSE and uncertainty error of the model significantly: MSE lowers from 0.1204 to 0.1161 and uncertainty error lowers from 0.0171 to 0.0119 (Table 2).

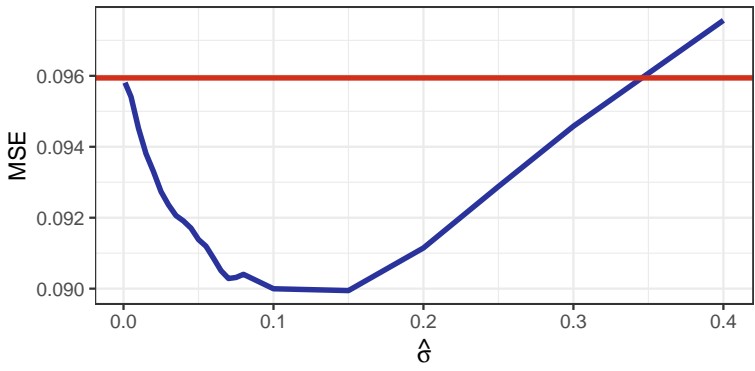

Figure 5: The MSE values with varying values of parameter $\hat{\sigma}$ in the simulation scenario with five-dimensional input. The red line refers to the prediction error without WTTA.

|  | Random forest | Random forest + WTTA |
|---|---|---|
| MSE | 0.1204 | 0.1161 |
| Uncertainty error | 0.0171 | 0.0119 |

Table 2: Performance of random forest and random forest with WTTA in the simulation scenario with five-dimensional input.

### 4.2 Wine classification study with multinomial logistic regression

In this study, we analyze a data set containing a chemical analysis of wine from three different cultivars Aeberhard & Forina (1992). This tabular data set contains 13 explanatory variables and one categorical response variable that refers to the wine cultivar. The data size is relatively small (n = 178) and the data is further divided into a training set (120 observations) and a test set (58 observations). All the 13 explanatory variables are continuous with varying scales. Before fitting a model, we standardize the variables with their z-scores to make it straightforward to utilize simple augmentation with a single scale parameter.

Next, we fit a multinomial logistic regression model to the training set. To implement the model, we use the r-package *nnet* and its function *multinom* Venables & Ripley (2002). We implement a k-fold cross-validation method with $k = 5$ to optimize the model hyperparameter $\hat{\sigma}$ using the r-package *modelr* Wickham (2022).

- Repeat the following process for each $\hat{\sigma} \in \{0.05, 0.1, 0.15, \dots 0.5\}$

- For each of the $k$ folds we

    1. Train the model with the remaining $k - 1$ folds.
    2. Apply the WTTA procedure for the selected fold with a single value for $\hat{\sigma}$.
    3. Evaluate the performance of the model via MSE

- Calculate the average performance over all folds

- The optimal value for $\hat{\sigma}$ is selected via the smallest MSE

In Figure 6 we observe the optimal value for parameter $\hat{\sigma} = 0.25$. Moreover, we can see that WTTA improves the results, in terms of prediction error, approximately when $\hat{\sigma} < 0.44$.

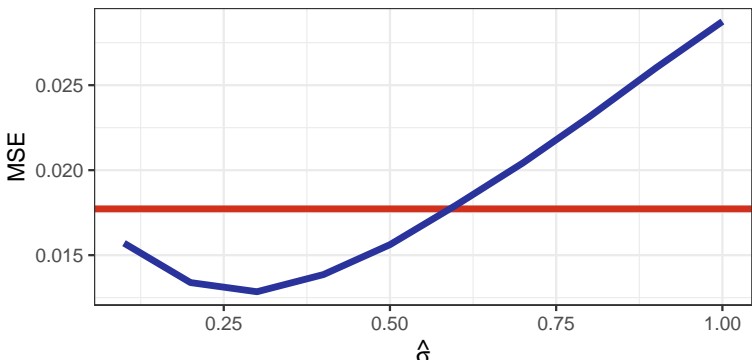

Figure 6: The MSE values with varying values of parameter $\hat{\sigma}$ over 5-fold cross validation in the wine classification scenario. The red line refers to the prediction error without WTTA.

### 4.3 Image classification study with neural networks

In this study, we examine both the standard TTA and the WTTA method with an image data set. We select the MNIST data set, which consists of black and white images of handwritten digits with the corresponding classes Deng (2012). The size of each image is 28x28 pixels. The training set of the data consists of 60000 images and the testing set has 10000 images, providing a relatively sizeable number of observations.

Generally, *neural networks* (NN) are the most suitable models to implement with such image data with their overwhelming prediction accuracy. In this study, we implement a general sequential NN model in R using Keras Allaire & Chollet (2022). The NN consists of three hidden layers with 256, 128, and 64 neurons. The last layer has 10 outputs, corresponding to the digit classes 0-9.

Implementation of the TTA method with image data in R is carried out with image transformation tools provided by the OpenImageR package Mouselimis (2023).

In addition, comparisons are made to existing calibration methods. Here, temperature scaling is used as a post-processing method to tackle the problem of overconfidence, where the final activation softmax function is rescaled with parameter $T$:

$$\text{softmax}(z) = \frac{e^{z/T}}{\sum_i e^{z_i/T}},$$

where $z$ are logits of the final layer.

In contrast to previous examples in this paper, where tabular data was utilized, MNIST data set is an image data set, where each variable corresponds to a value for each pixel. Previous examples of augmentation presented in this paper are not applicable as the neighboring pixels are highly dependent. Next, we have constructed an augmentation formula based on existing methods on images:

- Two tuning parameters are defined: $a_{\max}$ and $s_{\max}$ corresponding to maximum angle of rotation and maximum shift to the position of the image.

- Set values for $a_{\text{rand}}$ from uniform distribution $\text{Unif}(-a_{\max}, a_{\max})$ and values for $s_x$ and $s_y$ from $\text{Unif}(-s_{\max}, s_{\max})$.

- Generate new image by first rotating the image by an angle of $a_{\text{rand}}$ and shifting the image in x-coordinate by $s_x$ and y-coordinate by $s_y$.

- Pixel-wise distances between the generated images and the original image are calculated. With weighted augmentations each augmentation has an associated weight that is the inverse of the corresponding distance value.

In this study, we set the parameters $a_{\max} = 10$ and $s_{\max} = 3$ in order to produce 100 realistic augmented observations for each original observations that stay within the image frame. Alternatively these values could be determined with k-fold cross-validation method similarly to previous examples. However, with such a large number of observations in the testing set, this would result in high computation time.

In Table 3 we can observe that TTA and WTTA, when compared to the ordinary model, perform poorly with mean squared error. However, the total number of incorrect predictions goes down with WTTA.

Furthermore, various models with different sizes of training data are examined to explore the impact of training data to the prediction accuracy. Five sets of data are sampled from the original training data set with sizes 5000, 10000, 20000 and 30000, and with each set a similar NN model is fitted.

The MSEs of varying size of training sets are compared in Figure 7. We can observe that lowering the size of training data set worsens the MSE of each approach as expected, but weighted TTA performs better than the ordinary prediction with sizes 5000 and 10000.

Finally, TTA methods are examined in thresholded environment. We set a threshold raster from 1 to 0.6 by increments of 0.02 and analyze MSE and accuracy in each instance. Here the TTA and WTTA are compared to temporal scaling methods, named as Temp1, Temp2 and Temp3 with varying value for parameter $T$ (see Figures 8 and 9). We can clearly observe, that TTA and WTTA are better calibrated than Standard method. The calibrated predictions of the temperature scaling methods are in-between these two, providing some benefits to the standard approach.

## 5 Discussion and conclusion

In this paper, we presented the weighted test-time augmentation method for uncertainty quantification with various data sets and models. The method has several merits, as exemplified by our study scenarios: WTTA

|                      | NN     | NN + TTA | NN + WTTA |
|----------------------|--------|----------|-----------|
| MSE                  | 0.0268 | 0.0694   | 0.0375    |
| Classification error | 3.30%  | 3.54%    | 3.26%     |

Table 3: Performance of neural network, NN with TTA and NN with WTTA in the image classification scenario.

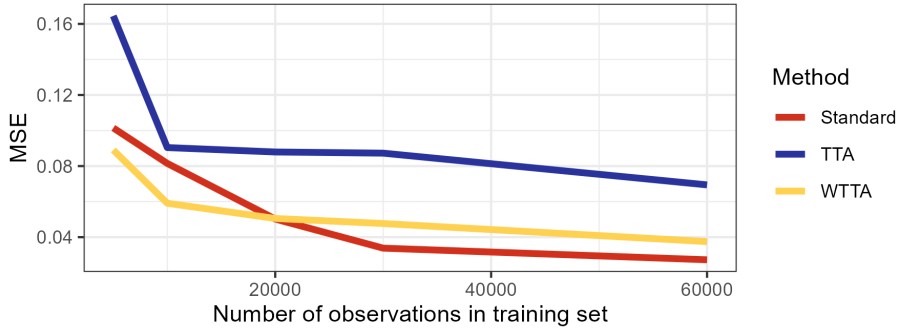

Figure 7: The MSE values with varying size of training sets in the image classification scenario.

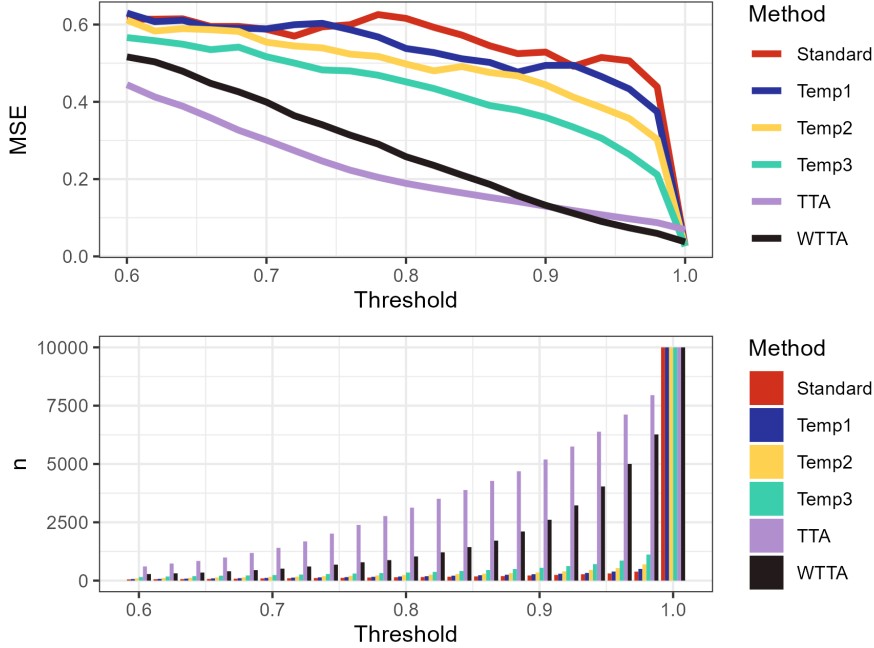

Figure 8: The thresholded values of MSE with different methods over full training data set in the image classification scenario.

improves uncertainty prediction as well as prediction accuracy in most cases, it can be implemented for all types of prediction models, and it does not require the original training data.

First, a simulation study was carried out to present the method with an easy-to-follow example, where comparisons to the "true" uncertainty could be made. In terms of both prediction and uncertainty error we were able to show that WTTA improves on ordinary model prediction.

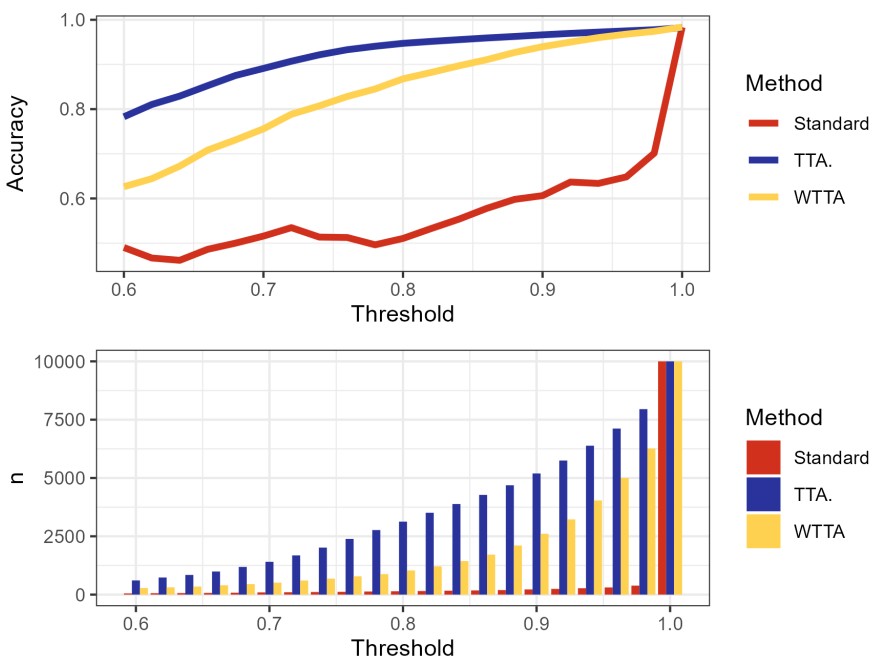

Figure 9: The thresholded values of accuracy with different methods over full training data set in the image classification scenario.

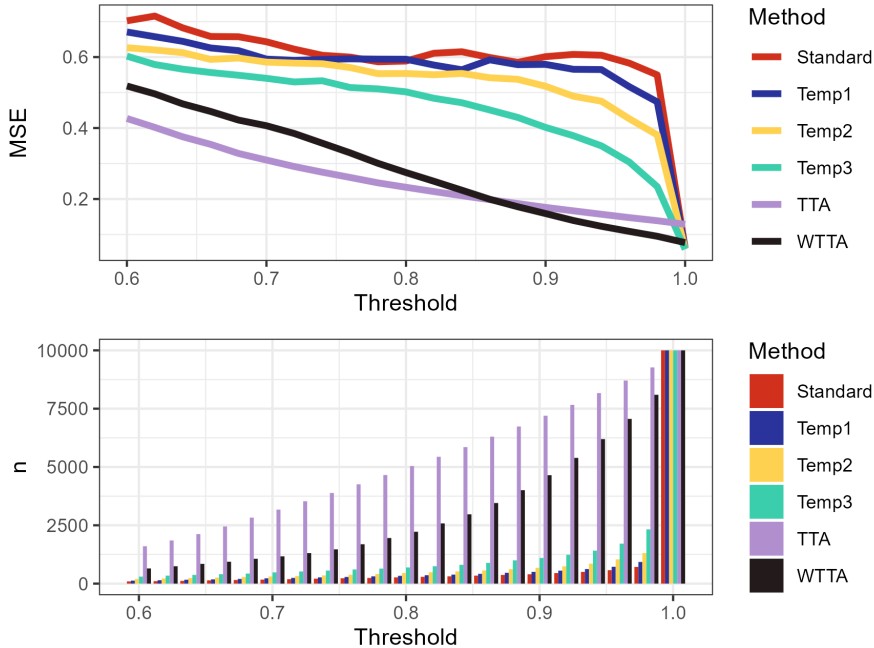

Figure 10: The thresholded values of MSE with different methods over training data set of size 10000 in the image classification scenario.

Second, a wine classification study was examined. In this example, we approximated the optimal value of the WTTA deviation parameter $\hat{\sigma}$ with 5-fold cross-validation. We showed that WTTA outperforms standard prediction, and that an optimal value for $\hat{\sigma}$ could be found.

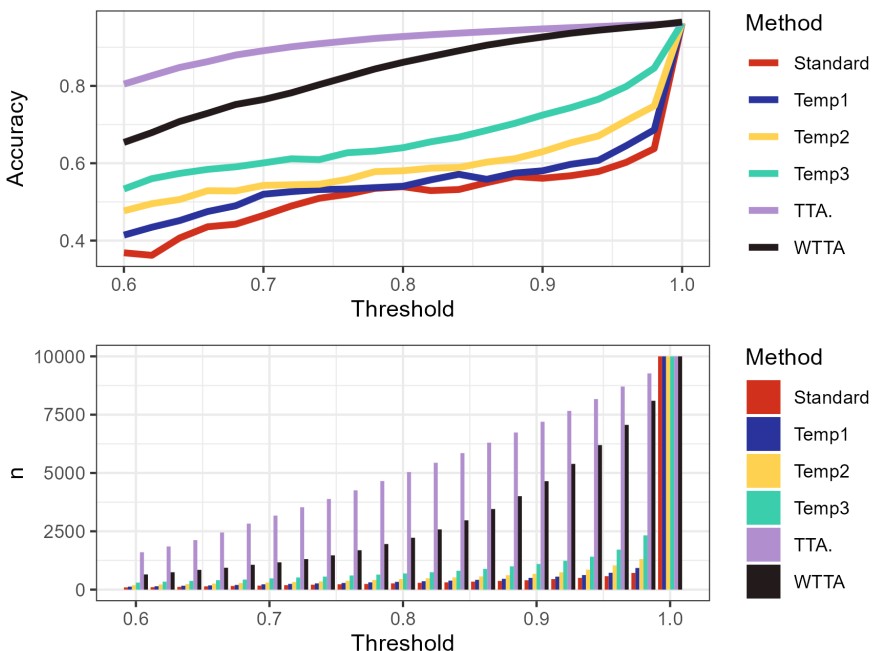

Figure 11: Thresholded values of accuracy with different methods over training data set of size 10000 in the image classification scenario.

Third, the commonly used data set MNIST is selected as an example, as it consists of image data. With images, the augmentation process of WTTA is carried out by randomly shifting and rotating the images and by using weights for each augmented observation. In terms of MSE, nonaugmented prediction turned out to perform better than the WTTA. However, the total number of incorrect predictions was slightly lower with WTTA. Morever, different sizes of training data were examined to explore the impact of the amount of training data to the WTTA. It was found that the lower the size of the training set, the better the WTTA method performed in terms of MSE.

In conclusion, WTTA proved to be useful method for uncertainty quantification with relatively low effort. The method was most effective with small data sets, such as in examples 1 and 2. This seems sensible as the augmentation process is similar to generating more data. Although with the MNIST data set, the WTTA was slightly worse than the ordinary prediction method in terms of MSE, the number of incorrect predictions was lower. This, together with confusion matrices, seems to indicate that WTTA predicts the borderline cases better than the ordinary prediction. However, the augmentation process always adds a certain amount of noise to the prediction that appears in the MSE, thus producing worse MSE overall. Still, the WTTA method would benefit from further research, especially in setting the optimal transformation function and defining the weights for each augmentation.

While the current approach of utilizing Gaussian distribution provides valid results in most of the cases, studying the impact of other alternatives could provide even more robust uncertainty measures.

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

## Appendix

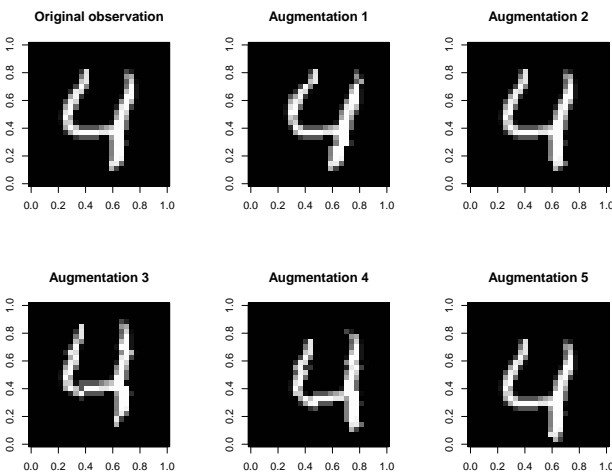

Figure 12: Example outcome images of the augmentation procedure.

|   | 0 | 1 | 2 | 3 | 4 | 5 | 6 | 7 | 8 | 9 |
|---|---|---|---|---|---|---|---|---|---|---|
| 0 | 971 | 1 | 0 | 1 | 0 | 1 | 1 | 1 | 3 | 1 |
| 1 | 0 | 1128 | 3 | 0 | 0 | 1 | 0 | 0 | 3 | 0 |
| 2 | 2 | 1 | 1017 | 1 | 1 | 0 | 1 | 4 | 5 | 0 |
| 3 | 0 | 1 | 3 | 986 | 0 | 7 | 0 | 6 | 3 | 4 |
| 4 | 0 | 1 | 3 | 0 | 965 | 0 | 4 | 1 | 1 | 7 |
| 5 | 1 | 0 | 0 | 8 | 1 | 875 | 2 | 1 | 2 | 2 |
| 6 | 5 | 2 | 1 | 0 | 1 | 2 | 946 | 0 | 1 | 0 |
| 7 | 0 | 0 | 6 | 1 | 1 | 0 | 0 | 1016 | 2 | 2 |
| 8 | 2 | 0 | 3 | 4 | 3 | 3 | 2 | 4 | 950 | 3 |
| 9 | 3 | 2 | 0 | 3 | 8 | 3 | 1 | 5 | 1 | 983 |

Table 4: Confusion matrix with ordinary prediction with training set a size of 60000.

|   | 0 | 1 | 2 | 3 | 4 | 5 | 6 | 7 | 8 | 9 |
|---|---|---|---|---|---|---|---|---|---|---|
| 0 | 974 | 1 | 0 | 0 | 0 | 2 | 1 | 1 | 1 | 0 |
| 1 | 0 | 1124 | 2 | 3 | 0 | 0 | 3 | 0 | 3 | 0 |
| 2 | 4 | 0 | 1017 | 1 | 1 | 0 | 1 | 6 | 2 | 0 |
| 3 | 0 | 0 | 1 | 1002 | 0 | 0 | 0 | 4 | 2 | 1 |
| 4 | 2 | 1 | 2 | 1 | 955 | 0 | 4 | 3 | 2 | 12 |
| 5 | 2 | 0 | 0 | 7 | 1 | 878 | 1 | 1 | 1 | 1 |
| 6 | 4 | 2 | 0 | 1 | 1 | 5 | 945 | 0 | 0 | 0 |
| 7 | 1 | 1 | 9 | 4 | 0 | 0 | 0 | 1007 | 2 | 4 |
| 8 | 6 | 1 | 2 | 4 | 1 | 5 | 0 | 3 | 950 | 2 |
| 9 | 4 | 3 | 0 | 4 | 2 | 2 | 2 | 5 | 2 | 985 |

Table 5: Confusion matrix with weighted augmented prediction with training set a size of 60000.

|   | 0 | 1 | 2 | 3 | 4 | 5 | 6 | 7 | 8 | 9 |
|---|---|---|---|---|---|---|---|---|---|---|
| 0 | 965 | 0 | 0 | 0 | 0 | 4 | 5 | 2 | 3 | 1 |
| 1 | 0 | 1123 | 3 | 1 | 0 | 1 | 5 | 0 | 2 | 0 |
| 2 | 7 | 1 | 965 | 12 | 3 | 2 | 11 | 11 | 19 | 1 |
| 3 | 0 | 0 | 3 | 973 | 0 | 24 | 0 | 6 | 3 | 1 |
| 4 | 1 | 1 | 2 | 0 | 932 | 3 | 17 | 3 | 6 | 17 |
| 5 | 5 | 1 | 0 | 7 | 0 | 867 | 4 | 0 | 6 | 2 |
| 6 | 7 | 3 | 0 | 0 | 5 | 13 | 928 | 0 | 2 | 0 |
| 7 | 2 | 6 | 6 | 3 | 1 | 2 | 1 | 998 | 2 | 7 |
| 8 | 4 | 1 | 1 | 6 | 3 | 14 | 5 | 3 | 935 | 2 |
| 9 | 4 | 8 | 1 | 16 | 19 | 9 | 2 | 4 | 7 | 939 |

Table 6: Confusion matrix with ordinary prediction with training set a size of 10000.

|   | 0 | 1 | 2 | 3 | 4 | 5 | 6 | 7 | 8 | 9 |
|---|---|---|---|---|---|---|---|---|---|---|
| 0 | 972 | 1 | 0 | 1 | 1 | 0 | 2 | 1 | 1 | 1 |
| 1 | 0 | 1125 | 1 | 2 | 0 | 1 | 2 | 0 | 4 | 0 |
| 2 | 0 | 0 | 1018 | 3 | 1 | 0 | 0 | 6 | 3 | 1 |
| 3 | 0 | 0 | 1 | 1001 | 0 | 2 | 0 | 4 | 1 | 1 |
| 4 | 1 | 1 | 5 | 0 | 952 | 0 | 4 | 2 | 2 | 15 |
| 5 | 2 | 0 | 0 | 7 | 1 | 875 | 2 | 1 | 3 | 1 |
| 6 | 3 | 2 | 0 | 0 | 2 | 2 | 949 | 0 | 0 | 0 |
| 7 | 1 | 3 | 10 | 3 | 0 | 0 | 0 | 1004 | 2 | 5 |
| 8 | 0 | 0 | 1 | 9 | 1 | 3 | 2 | 2 | 951 | 5 |
| 9 | 2 | 3 | 0 | 10 | 6 | 0 | 0 | 4 | 1 | 983 |

Table 7: Confusion matrix with weighted augmented prediction with training set a size of 10000.

