# OpenReview forum: "Uncertainty estimation in classification via weighted test-time augmentation"
_TMLR — Rejected by TMLR_

### Review · Reviewer_cwoH · 2025-11-18

**Summary Of Contributions:**

Deep learning models achieve high performance, but their predictions tend to be overconfident without proper uncertainty quantification, which limits their usability in safety-critical applications. This paper proposes a weighted version of test-time augmentation (WTTA), which incorporates distance-based weights when generating test-time augmentations. WTTA creates augmented samples using a Gaussian distribution and aggregates their predictions using a weighted mean. The authors evaluate WTTA on two-dimensional and five-dimensional datasets, as well as on the UCI Wine and MNIST datasets. Across all experiments, models equipped with WTTA achieve lower uncertainty error compared to standard predictions.

**Additional Comments:**

No comments.

**Audience:**

Yes

**Audience Explanation:**

Yes. Uncertainty quantification is a central topic in trustworthy machine learning, and WTTA provides a simple, model-agnostic post-hoc approach that can be applied to a wide range of classifiers. The paper evaluates the method across both tabular and image domains, offering practical insights that may benefit researchers working on reliability, calibration, and robustness.

**Broader Impact Concerns:**

No concerns.

**Claims And Evidence:**

No

**Claims Explanation:**

Strengths
* The proposed WTTA is a post-hoc method that can be combined with any model regardless of its architecture.
* WTTA reduces uncertainty error by introducing a simple weighting mechanism into the standard TTA procedure.

Weaknesses
* When applied to the image dataset, WTTA increases the MSE loss compared to the original model, unlike in other datasets.
* The experiments do not include more complex models or large-scale datasets; the paper only presents results using relatively simple models and small datasets.
* The underlying reason why WTTA improves uncertainty estimation compared to standard TTA is not sufficiently explained or theoretically analyzed.
* There is no analysis of how WTTA behaves under noisy or unrealistic augmentations, which is important for understanding its robustness.

**Requested Changes:**

1. Include an analysis of the conditions when WTTA succeeds and fails. Providing a theoretical or statistical justification would strengthen the persuasiveness of your argument.
2. Incorporate experiments using more complex models and large-scale datasets. Limiting the evaluation to simple models and small datasets reduces the generalizability of the proposed method.
3. An analysis of the behavior under noisy or unrealistic augmentations would help readers understand potential failure cases and important considerations when applying the method.
4. The paper claims that WTTA is robust to out-of-distribution (OOD) data, but the current experiments do not use clear OOD benchmarks. Please either justify that your benchmark represents an OOD scenario or include experiments using OOD datasets.
5. WTTA appears to introduce significant computational overhead, yet this is not sufficiently discussed. Please provide an analysis of the computational cost and compare it against standard TTA or other augmentation-based uncertainty estimation methods.

---

### Review · Reviewer_h8W1 · 2025-12-11

**Summary Of Contributions:**

Authors provide a weighted version of test time augmentation (WTTA) approach, with an idea to give weights to predictions for every different augmentation of the same data based on L2 distance. Authors proceed to show experiments on synthetic dataset and on two read world datasets (1 tabular and 1 image). The authors show that WTTA does perform better than TTA in all these settings. They also compare WTTA with temperature scaling methods in terms of calibration error and prediction accuracy.

However, I found the novelty to be very limited. And the experiments to be extremely limited without proper comparisons.

**Audience:**

No

**Audience Explanation:**

I find the contribution of the paper incremental and I don't think TMLR's audience will be interested in it.

**Broader Impact Concerns:**

Not required

**Claims And Evidence:**

Yes

**Claims Explanation:**

Yes the paper has experiments which looks accurate, clear and convincing.

**Requested Changes:**

# Weakness
- The paper has very limited novelty as it suggests to use a weighted aggregation of predictions across different augmentations of a data point. Moreover, it only suggests a very simple L2 distance based metric to do so.
- The weight calculation only allows and makes sense for continuous variables, whereas tabular data often has discrete variables and these types of addition does not makes sense in those settings.
- The experiments with synthetic dataset seems biased towards WTTA approach. I say this because if I have a dataset with 2 dimensional features and suppose 2nd dim is much more sensitive to flip the label with small perturbation. Ideally more weight should be given for an $\epsilon$ change in 2nd dim vs an $\epsilon$ change in 1st dim. However, L2 based weight will give same weight in both settings.
- Even for experiment with images there is not motivation of why the suggested weights makes sense, ideally I would assume the weights should be based on $a_{rand}, s_x, s_y$.
- There are many other TTA methods which needs comparison.
- **Also the paper authors cite [1] already introduces generalized TTA which is TTA with weights.** Not only this there are other works which suggests different weighted TTA approaches [2].

[1] Kimura, M. (2024). Understanding Test-Time Augmentation. ArXiv, abs/2402.06892.
[2] Kimura, M., & Bondell, H.D. (2024). Test-Time Augmentation Meets Variational Bayes. ArXiv, abs/2409.12587.

---

### Review · Reviewer_g5EW · 2025-12-16

**Summary Of Contributions:**

This work proposes Weighted Test-Time Augmentation (WTTA) as a model-agnostic approach to quantify uncertainty in model predictions. It utilizes the variance in prediction among multiple augmented samples to estimate the model uncertainty. In particular, the method proposes a distance-based weighting scheme to the augmented samples, where samples closer to the original input are given higher weights in uncertainty estimation. This method does not requires training or architectural changes, and is easy to apply during test time on top of existing models.

**Audience:**

Yes

**Audience Explanation:**

1. The work addresses an important and interesting topic of uncertainty quantification. It helps to target overconfidence in model predictions, which is a key limitation in many applications.
2. The proposed method is a simple model-agnostic technique, applicable without heavy modifications.

**Broader Impact Concerns:**

No broader impact concerns

**Claims And Evidence:**

No

**Claims Explanation:**

The work could be strengthened with the following points:

1. **Limited experiment scope**: The experiments are restricted to small and simple datasets (eg. synthetic data, MNIST, and wine classification). It remains unclear how well the proposed method and its conclusions would generalize to large-scale datasets that are more representative of current research and real-world deployment.
2. **Insufficient justification**: The proposed weighting scheme based on distance between original and augmented samples seems heuristic. Providing more theoretical motivation or analysis for this design choice would help strengthen the work.
3. **Additional ablation studies**: following (2), comparing the proposed weighting with other weighting strategies would help demonstrate its effectiveness. For example, entropy-based weighting, learned weights, etc might be common & good starting points.
4. **Additional baselines**: The evaluation only compares against standard TTA and temperature scaling under different temperatures, but lacks comparison with recent works on uncertainty quantification/calibration.

**Requested Changes:**

My main concerns are listed above; some minor comments/questions:

1. The overall presentation could be improved by: (a) more clearly stating the proposed method, its scope, and baselines compared to, and (b) organizing experimental results more coherently (e.g., grouping related visualizations and tables together), rather than interleaving them with the main text which may sometimes be obstructing.
2. How does the approach using sample augmentations compare to uncertainty estimated by model ensembles, where the variance stems from different models instead of different augmentations?

---

### Author Response · Authors · 2026-01-02
**Response to comments**

We thank all the reviewers for their comments. We found those very constructive and useful for the paper. Here is our response to all the comments as there were many similarities between requested changes. Please note that as this review process was during the holiday season, we had a limited time to make changes to the manuscript. Now in January, we can start to implement these changes to the manuscript.

First, more datasets and models were requested to provide statistical justification for the WTTA method. We find this recommendation reasonable and relatively easy to implement. We agree that the current scenarios are quite simple and large scale dataset would make our argumentation better. Furthermore, more complex models could be implemented with WTTA. Especially, we would like to thank reviewer cwoH for the comment about out-of-distribution data and benchmarks as this type of experiments would clearly strenghten the work. Also, comment from reviewer h8W1 regarding that the approach makes sense only with continuous variables is reasonable and we should discuss that more. They also argue that the simulated data set could be improved, which we totally agree.

Second, the request to make more comparisons to other methods is brought up and reviewer h8W1 kindly provides citations of similar test-time augmentation methods. We agree that more comparisons could help the paper to further justify the method. These include other TTA methods, different weighting strategies and other calibration methods than temperature scaling. The reason for not including these comparisons in the current manuscript is that the proposed WTTA method is somewhat unique when it comes to the models and data. WTTA is a model agnostic technique, so comparisons to model specific uncertainty approaches would be unfair to some extent. Additionally, WTTA method does not require the original training data set to quantify uncertainty and is therefore rather simple to apply to different scenarios. This differs from most other weighted TTA approaches, but is not properly discussed in the current version of the paper. However, we find that some comparisons to existing methods, especially to other TTA methods, should be made to at least show the different requirements of the methods and possible drawbacks.

Third, the overall presentation of the paper was mentioned. We agree that with so many figures and tables the readability suffers a little and we will organize those better. Moreover, the presentation of the proposed method could be improved as well, making it clear what are the requirements and scope of the method and what are aim of the comparisons presented.

To conclude, we aim to improve the paper by

1. adding a new example with a large-scale dataset

2. adding a new OOD example

3. implementing more complex models to these scenarios

4. modifying the current simulated scenario

5. implementing an alternative TTA approach for comparison

6. clearly depicting the requirements of the proposed method and the possible drawbacks when compared to the alternatives

7. reorganizing the model description, figures and tables to increase readability

8. discussing the computational burden of the proposed method

---

### Decision · Action_Editor_7Pej · 2026-02-22

**Recommendation:** Reject

**Audience:**

Yes

**Audience Explanation:**

It is interesting to the part of the community interested in uncertainty in LLMs

**Claims And Evidence:**

No

**Claims Explanation:**

It appears that in the current version of the paper the scope of the experiments is quite narrow, and more baselines and models need to be added to support the claims. I would encourage the authors to follow up on the comments by the reviewers, and improve the paper by adding newer experiments.

**Resubmission Of Major Revision:**

The authors may consider submitting a major revision at a later time.